# STED Direct Laser Writing of 45 nm Width Nanowire

**DOI:** 10.3390/mi10110726

**Published:** 2019-10-28

**Authors:** Xiaolong He, Tianlong Li, Jia Zhang, Zhenlong Wang

**Affiliations:** 1Key Laboratory of Micro-systems and Micro-structures Manufacturing, Ministry of Education, Harbin Institute of Technology, Harbin 150080, China; hexiaolong8586@163.com (X.H.); tianlongli@hit.edu.cn (T.L.); zhangjia@hit.edu.cn (J.Z.); 2School of Mechatronics Engineering, Harbin Institute of Technology, Harbin 150001, China; 3Institute of Pharmacy, Sechenov University, 119991 Moscow, Russia

**Keywords:** controlled fabrication, 45 nm width, STED direct laser writing, rod-shape effective focus

## Abstract

Controlled fabrication of 45 nm width nanowire using simulated emission depletion (STED) direct laser writing with a rod-shape effective focus spot is presented. In conventional STED direct laser writing, normally a donut-shaped depletion focus is used, and the minimum linewidth is restricted to 55 nm. In this work, we push this limit to sub-50 nm dimension with a rod-shape effective focus spot, which is the combination of a Gaussian excitation focus and twin-oval depletion focus. Effects of photoinitiator type, excitation laser power, and depletion laser power on the width of the nanowire are explored, respectively. Single nanowire with 45 nm width is obtained, which is λ/18 of excitation wavelength and the minimum linewidth in pentaerythritol triacrylate (PETA) photoresist. Our result accelerates the progress of achievable linewidth reduction in STED direct laser writing.

## 1. Introduction

The emergence of stimulated emission depletion (STED) microscopy has broken the diffraction barrier in optical imaging [1,2,3,4], and revolutionized the lateral resolutions down to 5.8 nm [5]. Clearly, this dramatic resolution enhancement in optical microscopy is highly expected to translate into a homologous improvement in optical lithography, especially in two-photon direct laser writing (DLW) [6,7,8]. In two-photon DLW, a femtosecond laser beam is tightly focused into a photo sensitive material, and only a tiny voxel around the beam waist can be sufficiently exposed through two-photon absorption [9,10]. By numerical control the scanning route of laser beam, 2D or 3D structures with arbitrary shape can be obtained, realizing 3D printing with sub-micron resolution [11,12]. Although these micro or sub-micro structures have found applications as photonic devices [13,14], biomedical components [15,16,17], microfluidic devices [18,19,20], and electronic circuits [7,21,22], structures with nanometer feature size will find wider applications. 

For STED direct laser writing, a second laser beam, also called a depletion laser beam, is employed to overlap with the first direct laser writing beam (excitation laser beam) and deplete previously excited photoinitiators [23,24,25,26]. Thus, the capability of the photoinitiator to polymerize photoresist molecules is suppressed. The depletion laser focus always features with a spatial shape, which includes a zero-intensity region at the location where the excitation focus has its maximum intensity [23,27,28]. With the increasing of depletion laser power, a growing number of photoresist molecules are forced into the non-polymerized state, with the exception of the molecules located at zero depletion intensity region [29]. Functional 2D and 3D sub-diffraction nanostructures, for example photoresist nanowires and nano brush, have been introduced by STED lithography, which are necessary for applications in photonics [30,31], electronics [32], and life science [33,34]. In conventional STED-DLW, normally a donut-shaped depletion focus is used to fabricate stable nanowires and 3D sub-diffraction nanostructures. Since the excitation focus is totally encircled by the depletion focus and excessive depletion power can also excite the photoresist molecules, the minimum width or feature size in lateral direction is restricted to 55 nm [30] and 53 nm [28,31]. Gan et al. use the gap between excitation and donut depletion focus in STED lithography, and achieve single and two adjacent lines with a 9 nm feature size [35]. While, these line structures are below 50 nm, they are quite short and not very stable. In this work, we used STED direct laser writing with a rod-shape effective focus spot to write stable and continuous nanowires below 50 nm. This rod-shape effective focus spot was a combination of the Gaussian excitation focus and twin-oval depletion focus. An fs-laser (800 nm Ti: sapphire) and 532 nm continuous wave laser were employed as the excitation and depletion source. To achieve the minimum linewidth, effects of the photoinitiator type, excitation power, and depletion power on the width of the nanowire were explored. A stable and continuous nanowire with 45 nm width was obtained, which was λ/18 of excitation wavelength and the thinnest nanowire in pentaerythritol triacrylate (PETA) photoresist. A series of nanowires with 45 nm width were fabricated to verify the stability of our method, and a higher depletion power was applied to explore the smaller nanowire width.

## 2. Experimental Details

### 2.1. Methods

The schematic diagram of STED direct laser writing method is shown in Figure 1a. Our experimental system consisted of three parts, laser source setup, optical beam delivery components, piezo stage and its control system. For the laser source part, a Ti: sapphire femtosecond laser (Micra-10, Coherent Inc., Santa Clara, CA, USA) with 800 nm wavelength, 38 fs pulse duration and 80 MHz repetition rate was used as the excitation source, and a 532 nm continuous-wave (CW) laser (Millennia Pro, Spectra-Physics Laser Inc., CA, USA) was used as the depletion source. Both of the two laser beams were employed in a Gaussian shape mode. Two telescopes were used to enlarge the diameter of 800 nm and 532 nm wavelength laser beam for fulfilling the numerical aperture (NA) of the objective lens, which focused the laser beam into the sample photoresist. A half-space edge phase mask was inserted into the depletion laser beam (532 nm) to modify its focal spot to the twin-oval mode as shown in Figure 1b. The twin-oval depletion beam had two oval-shape high intensity areas, which were separated by a zero-intensity valley in the beam center. After the objective lens, the Gaussian shape excitation focus coincided with twin-oval depletion focus, resulting in a sub-diffraction feature size rod-shaped effective focus spot as shown in Figure 1c. A beam splitter was inserted into both the two laser beams before the objective lens to make them linearly polarized, and the 100× oil immersion objective lens (Apo TIRE, Nikon Co., Tokyo, Japan) had a NA of 1.49. The sample photoresist was drop-cast onto a 200 µm thickness cover slice (12-540A, Fisherbrand, MA, USA), which was fixed on top of a three-axis piezo-electric stage (P563, Physik Instrumente Co., Karlsruhe, Germany) mounted on an optical microscope (Eclipse Ti-U, Nikon, Tokyo, Japan). A refractive index match between immersion oil, cover slice, and the photoresist was conducted to reduce the aberrations. A charge-coupled device (CCD) camera (INFINITY1-2CB, Lumenera Co., Ottawa, Canada) was used to observe the state of focus spot and monitor the nanostructure fabrication process.

### 2.2. Materials

For our STED direct laser writing experiments, a monomer PETA was used as the sample photoresist. This monomer contained 300–400 ppm monomethyl ether hydroquinone as the photoinhibitor. This photoinhibitor can reduce the automatic coagulation of the photoresist, and also deters the photochemical reaction. To accelerate the excitation and depletion process, a photoinitiator should be added into this monomer. The type of photoinitiator is one of the main factors influencing the photon absorption, fluorescence emission, and eventually has effects on the width of the nanowire. In previous studies, normally two photosensitizer are used. One is 7-diethylamino-3-thenoylcoumarin, DETC for short and the other is isopropyl thioxanthone (ITX) [24,29]. To investigate the effects of these two photoinitiators in STED direct laser writing, 0.01 g DETC and 0.01 g ITX were separately added into two bottles with 1.99 g PETA photoresist inside, and no further chemicals were added. To thoroughly mix the photoresists, the two bottles were stirred for 20 min. In this way, 0.5 wt% ITX in PETA and 0.5 wt% DETC in PETA were obtained. Then, we measured the absorbance spectra of DETC and ITX using a UV-VIS spectrophotometer (Varian Cary 60 Scan) and the fluorescence spectra with fluorescence spectrophotometer (Varian Cary Eclipse), as shown in Figure 2. The 0.5 wt% DETC in PETA showed a peak fluorescence emission at 492 nm wavelength, and falling to 54.3% at 532 nm. The 0.5 wt% ITX in PETA showed a peak fluorescence emission 438 nm wavelength, and falling to 4.9% at 532 nm. For photoinitiators showing high two-photon excitation at 800 nm wavelength, an identical one-photon absorption at 400 nm can also be found. DETC and ITX showed a similar one-photon absorption at 400 nm wavelength, which was 58.2% and 60.5%, while the former fluorescence emission at 532 nm was about ten times larger than the latter. It meant that ITX in PETA needed much more depletion power to meet the same result with DETC in PETA. Nevertheless, the high depletion laser may have resulted in wider nanowires due to the residual two-photon absorption of 532 nm CW laser [36].

### 2.3. Characterization

After exposure, the samples were developed in isopropanol for 10 min, and then rinsed with acetone for 1 min. After that, 10 nm of gold was evaporated on the samples surface to make them conductive. A field emission scanning electron microscope (Hitachi S-4800, Hitachi Ltd., Tokyo, Japan) was used to observe the surface topography of the samples. The width of the nanowires were measured with the combination of imaging software and SEM images. An atomic force microscope (Auto probe CP, Thermo-Microscopes, MA, USA) was used to get the AFM images and height profile of nanostructures, and all scans were in non-contact mode. 

## 3. Results and Discussion

### 3.1. Effects of Photoinitiator

Straight nanowires were written to explore the effect of the photoinitiators. The nanowire in Figure 3a was written with 0.5 wt% ITX in PETA, and the nanowire in Figure 3b was written with 0.5 wt% DETC in PETA. Both of the two nanowires were written with a 15 mW excitation laser (red) power and 30 mW depletion laser (green) power. All the laser powers were measured before the objective lens. The objective had a transmittance of 70% for 800 nm wavelength, 84% for 532 nm wavelength. The focal spot diameter of the objective lens at the two wavelengths can be obtained from th eAbbe equation, which is 268 nm and 179 nm, respectively. The 800 nm wavelength excitation had a pulse duration of 38 fs and repetition rate of 80 MHz. Then, it was found the single pulse energy at the focal point was about 0.13 nJ, the laser fluence was about 0.23 J/cm^2^ and the peak intensity was 6.05 × 10^12^ W/cm^2^. For continuous wave depletion laser, the laser intensity was 1.003 × 10^8^ W/cm^2^. The scanning speed was 50 μm/s. The scanning direction was along the y direction, parallel to the laser polarization direction, and the depletion laser was switched on during writing. The one-beam direct laser writing nanowires (bottom half of nanowires) in Figure 3a,b had the same width, about 210 nm, since the two photoinitiators showed a similar one-photon absorption at 400 nm wavelength. For two-beam STED direct laser writing nanowire (top half of nanowire) in Figure 3a,b, there was a clearl line width reduction when the depletion laser was switched on. There was no size reduction when the writing direction was perpendicular to the laser polarization direction. These size reductions were mainly located at the sides of nanowires. This is because the lateral spatial polymerization was inhibited by the high depletion intensity. Only molecules in the zero-intensity valley, located at the center of the depletion beam, can be polymerized smoothly. The STED-DLW nanowire with 0.5 wt% DETC as photoinitiator had a width of 134 nm as shown in Figure 3b, which was much thinner than the nanowire with 0.5 wt% ITX in Figure 3a, 176 nm width. The STED direct laser writing results indicated that 0.5 wt% DETC in PETA showed a higher fluorescence emission efficiency at 532 nm wavelength than 0.5 wt% ITX in PETA. Therefore, DETC was chosen as the photoinitiator in the below STED direct laser writing combining the experimental results and spectra analysis of Figure 2.

### 3.2. Effect of Excitation Power

The two-photon excitation power is a critical parameter influencing the minimum feature size attained with direct laser writing as well as STED direct laser writing. This time, we also briefly drew single nanowires. To achieve the minimum line width in STED direct laser writing, the thinnest nanowire in direct laser writing (red beam only) should be obtained first. The excitation laser power was continuously reduced from 15 mW, where we tested the two photoinitiators above. The scanning speed was unchanged at 50 μm/s, and the scanning direction was along y direction. Single nanowire writing with 15 mW excitation power had a width of 210 nm as shown in Figure 4a, consistent with the value we obtained in Figure 3. The thinnest nanowire was attained with 11 mW excitation power, which had a width of 120 nm as shown in Figure 4b. No nanowire was found with excitation power below 11 mW.

### 3.3. Effect of Depletion Power

The depletion power is another critical parameter influencing the minimum line width in STED direct laser writing. Since we obtained the thinnest nanowire in the one-beam direct laser writing experiment with 11 mW excitation power, line width reduction STED direct laser writing experiments were exemplified then. For the two-beam approach, the two-photon excitation power was set to be constant at 11 mW, the depletion laser (green) power was continuously increased from 10 mW. The scanning speed was 50 μm/s, scanning direction was parallel to the laser polarization direction, and the depletion laser was switched on during writing. The experimental results are shown in Figure 5. The one-beam direct laser writing nanowires (bottom half of the nanowires) had a width of 120 nm, showing no difference with the value in Figure 4b. For two-beam STED direct laser writing nanowires (top half of the nanowires in Figure 5), there was a clear width reduction when the depletion laser was switched on, and the linewidth became even smaller with the increasing depletion power. Finally, a minimum linewidth of 45 nm was reached at the exciting and depletion power of 11 and 25 mW respectively, as shown in Figure 5d. This number also consisted of the 10 nm thick gold film for SEM imaging. These linewidths were averaged by a series of (at least five) nanowires with the same parameters to reduce error. No nanowire was found when the depletion power came to 30 mW. Slowly increasing the depletion power from 25 mW may lead to a smaller line width, while the nanowires may not be stable or continuous. 

Atomic force microscope (AFM) scanning was performed to further characterize these nanowires with non-contact mode, as shown in Figure 6. Figure 6a,b shows the AFM image of the nanowires we obtained above, and Figure 6c shows the height profile of them. It was seen that the size reduction was in both the width and the height of these nanowires when the depletion laser was switched on. In Figure 6c, the one-beam DLW nanowire has a width of 160 nm, and the height is about 72 nm. The linewidth measured was greater than the value of observed in Figure 5, 120 nm. This was due to the AFM tip, which has a non-ignorable size. The value of 160 nm was the convolution of actual linewidth and AFM tip size. The two-beam STED-DLW nanowire had a height of 55 nm, 47 nm, 39 nm and 32 nm, respectively. This means that with the increasing of depletion power, the height and width of STED-DLW nanowires decreased simultaneously.

To further explore the effect of higher depletion power and verify the stability of our method, a series of nanowires were fabricated using STED direct laser writing experiment with 25 mW and higher depletion power. A 30 mW depletion power was used and the other parameters remained unchanged, the STED direct laser writing nanowires is shown in Figure 7a. The bottom half of nanowire written by one-beam DLW had a linewidth of 120 nm, which was stable and continuous as shown, while no line was found in the top half. The two-beam STED-DLW nanowire was totally removed due to the higher depletion power. One reason is that the thin nanowire collapsed and fractured, which means the photoresist could have been washed away during development. The other is the loss of the global connectivity between the polymer chains, resulting in the decomposition and fracture of the photoresist molecule within the nanowire. Too high depletion laser power will decompose the photoresist molecule and make nanowires unstable, disconnected, and eventually disappear. These experimental results validated that 45 nm was the minimum linewidth we obtained in STED direct laser writing experiments. Figure 7b shows a series of STED-DLW nanowires with 45 nm width fabricated with the parameters used in Figure 5d, and the length of these nanowires was about 1 μm.

## 4. Conclusions

In this work, we used STED direct laser writing with rod-shape effective focus spot for fabricating nanowires below 50 nm. This rod-shape effective focus spot was the combination of a Gaussian excitation focus and twin-oval depletion focus. A visible light 800 nm wavelength Ti: sapphire femtosecond laser and a 532 nm continuous wave laser were used as the excitation and depletion source. With a proper photoinitiator type, excitation power and depletion power, a single nanowire with 45 nm width was obtained in PETA photoresist, which is λ/18 of the excitation wavelength and the thinnest in this photoresist. A series of nanowires with 45 nm width were fabricated to verify the stability of our method, and a thinner nanowire could be achieved with a higher depletion power and monomer of higher crosslinking density. Our method, capable of writing stable and continuous nanowires below 50 nm, has accelerated the progress of precision enhancement in optical nanofabrication. 

## Figures and Tables

**Figure 1 micromachines-10-00726-f001:**
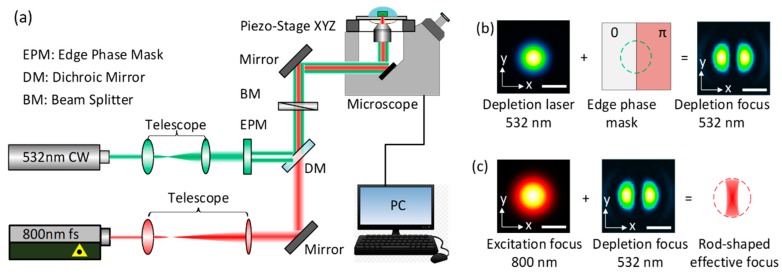
(**a**) Schematic diagram of simulated emission depletion (STED) direct laser writing. (**b**) Creation of the twin-oval depletion focus by inserting a half-space edge phase mask into the depletion beam. (**c**) Formation of rod-shaped effective focus spot, which was a combination of Gaussian excitation focus coinciding with twin-oval depletion focus, scale bar 200 nm.

**Figure 2 micromachines-10-00726-f002:**
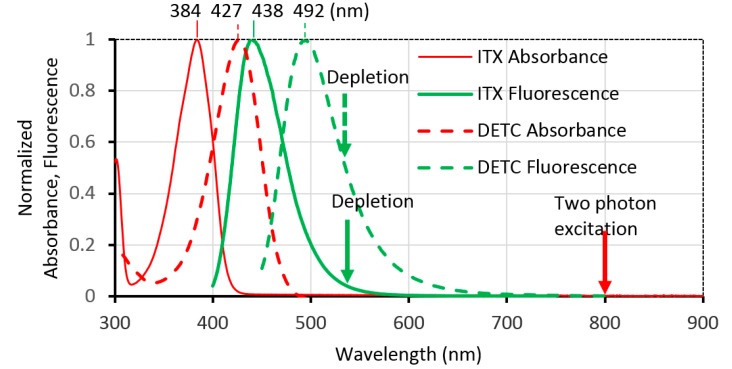
Absorbance and fluorescence spectra of two photoinitiators, solid line: 0.5 wt% isopropyl thioxanthone (ITX) in pentaerythritol triacrylate (PETA), dashed line: 0.5 wt% 7-diethylamino-3-thenoylcoumarin (DETC) in PETA. STED direct laser writing experiment was conducted to investigate the effect of the two photoinitiators.

**Figure 3 micromachines-10-00726-f003:**
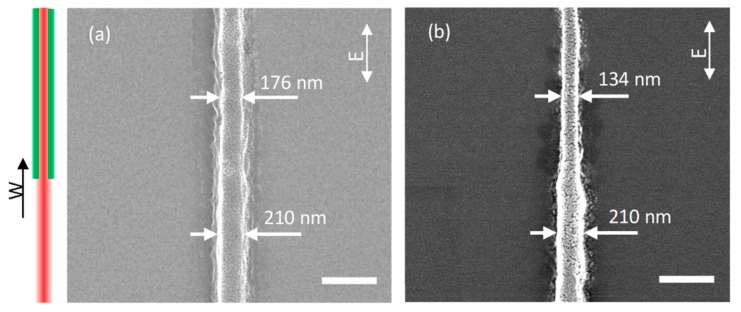
Scanning electron micrograph of nanowires written in PETA with two different photoinitiators, (**a**) 0.5 wt% ITX in PETA and (**b**) 0.5 wt% DETC in PETA. W: scanning direction, E: laser polarization direction, scale bar 400 nm.

**Figure 4 micromachines-10-00726-f004:**
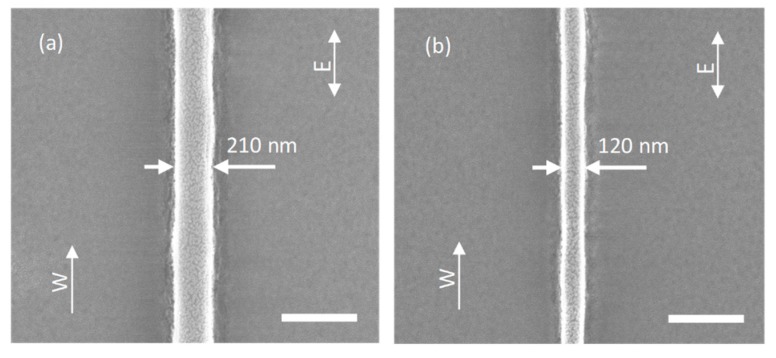
Scanning electron micrograph of one-beam direct laser writing nanowires, (**a**) 15 mW excitation power, (**b**) 11 mW excitation power. W: scanning direction, E: laser polarization direction, scale bar 400 nm.

**Figure 5 micromachines-10-00726-f005:**
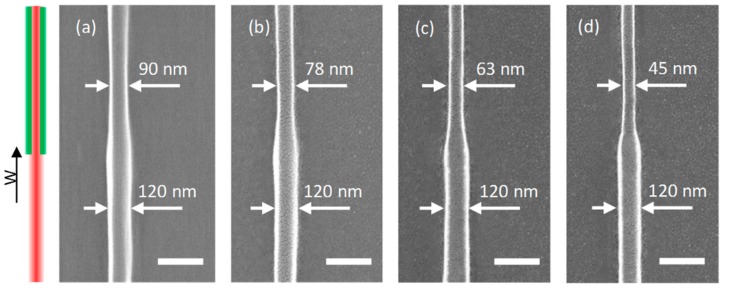
STED direct laser writing nanowires with 11 mW two-photon excitation power and various depletion powers, (**a**) 10 mW, (**b**) 15 mW, (**c**) 20 mW, (**d**) 25 mW depletion laser (green) power. The arrow W indicates the scanning direction, scale bar 200 nm.

**Figure 6 micromachines-10-00726-f006:**
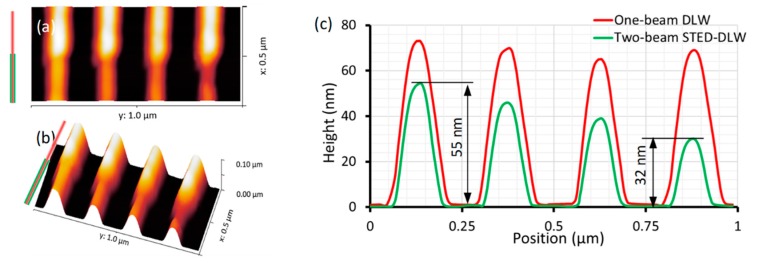
(**a**,**b**) Atomic force microscope (AFM) images of fabricated nanowires (**c**) height profile of the four nanowires, with 11 mW two-photon excitation power and depletion power from left to right 10, 15, 20, and 25 mW respectively.

**Figure 7 micromachines-10-00726-f007:**
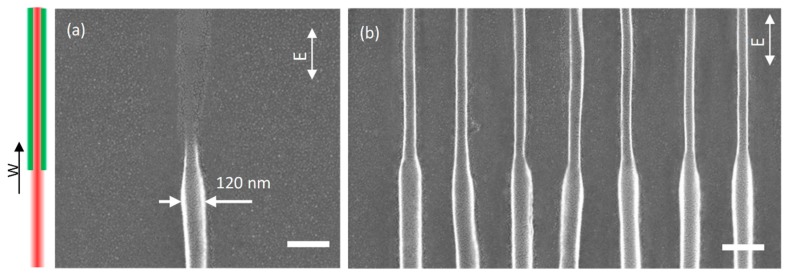
Scanning electron micrograph of STED direct laser writing nanowires. (**a**) 11 mW two-photon excitation power, 30 mW depletion power. (**b**) A series of STED-direct laser writing (DLW) nanowires with 45 nm width. W: scanning direction, E: laser polarization direction, scale bar 200 nm.

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
