# Peer review of "STED Direct Laser Writing of 45 nm Width Nanowire"

_micromachines, 2019, doi:10.3390/mi10110726_

Round 1
Reviewer 1 Report
This manuscript demonstrates a technique bases on STED to fabricate nanowires having a width of 45 nm.
The result is quite convinced but the technique itself is not new! Indeed, STED based direct laser writing technique has been demonstrated as un useful technique to realize sub-diffracted structures in 1D, 2D and 3D.
However, there are still a lot of work to do in order to make this STED-based DLW reliable and repeatable, and this manuscript tried to propose a way to do so. The idea of the authors is to use a twin-oval focusing spot to compress the laser spot in one direction, thus producing a narrow line when scanning the focusing spot in one direction.
First, we should mention that this idea has been demonstrated so far: ACS Nano 2011, 5, 5141, Chem. Phys. Chem 2012, 13, 1429, etc.
The authors claim that their results are more stable, but no any prove for this claim. I propose that the authors make a series of 1D nanowires with the same width of 45 nm by this technique. If statistically, all the nanowires are stable and continuous, the authors can claim such sentence.
Secondly, in abstract and conclusion, the authors mention that “We expect that our approach, in particular, the fabrication of nanowires below 50 nm could be a prospective fabrication method for nanostructured components and devices.” or “Our approach, capable of writing stable and continuous nanowires below 50 nm, could offer a promising high-resolution manufacturing method for nanoscale devices and components.”
Which nanoscale devices and components ?
Because, it is clear that this twin-oval focusing spot can only reduce the focusing spot side in one direction. By scanning in one direction (a straight line), we can obtain a narrow straight line. This is not possible to obtain a curved line or a 3D structure with same small size ! So, which nanoscale devices and components ? Only a series of straight nanowires can be realized by this technique.
The authors should remove these sentences.
Similarly, in the section 3, the authors write: “Here, we simply write straight nanowires instead of 3D nanostructures on the cover slice substrate, since the widths from single nanowires have a higher reliability than those feature size from 3D nanostructures.”
That sentence is not correct, because no 3D structure can be fabricated by this method ! as explained above.
The result shown in Figure 6 is not convinced. In figure 6a, I don’t see any reduction of the wire size when the STED beam in on (it is very clear in Figure 5). Simply a reduction of the height of the wire, as shown in Figure 6c. So the width is not reduced ? If the STED works for 30 mW of power (continue from the result of Figure 5h), we should first see a reduction of the wire width, even this “small wire” will be collapsed and washed out during the development process. Here, clearly see in Figure 6a, that the width is the same. The authors should check for the result at higher STED laser power and correct this result.
Finally, the authors should correct the term “one-photon” and “two-photon” because there are many times in the text, it is written as “two photon”.
The manuscript requires thus a major revision
Author Response
Responses to Reviewer’ Comments of micromachines-610158
The authors would like to thank the reviewer for the insightful comments. The following are detailed responses to the reviewer’s comments.
This manuscript demonstrates a technique bases on STED to fabricate nanowires having a width of 45 nm.
The result is quite convinced but the technique itself is not new! Indeed, STED based direct laser writing technique has been demonstrated as un useful technique to realize sub-diffracted structures in 1D, 2D and 3D.
However, there are still a lot of work to do in order to make this STED-based DLW reliable and repeatable, and this manuscript tried to propose a way to do so. The idea of the authors is to use a twin-oval focusing spot to compress the laser spot in one direction, thus producing a narrow line when scanning the focusing spot in one direction.
First, we should mention that this idea has been demonstrated so far: ACS Nano 2011, 5, 5141, Chem. Phys. Chem 2012, 13, 1429, etc.
The authors claim that their results are more stable, but no any prove for this claim. I propose that the authors make a series of 1D nanowires with the same width of 45 nm by this technique. If statistically, all the nanowires are stable and continuous, the authors can claim such sentence.
Response: We thank the reviewer for bringing these two papers to our attention and agree that the twin-oval focus have been demonstrated in several previous work. While, Stöhr et al. (Ref.1 below) use twin-oval focusing spot for imaging and patterning graphene, and the thinnest nanowire obtained by Harke et al. and Wegener et al. (Ref. 2 and Ref. 3) is 60 nm width. In this work, we focus on fabrication nanowires below 50 nm, the thinnest nanowire we acquired has a linewidth of 45 nm, much thinner than previous work.
As the suggestion of the reviewer, an image of a series of stable nanowires with 45 nm width has been added to the revised manuscript in Figure 7b. A line to describe the details of this figure has been added in p. 6, as: “Figure 7b is a series of STED-DLW nanowires with 45 nm width, and the length of these nanowires is about 1 μm.”
Figure 7. Scanning electron micrograph of STED direct laser writing nanowires. (a) 11mW two-photon excitation power, 30 mW depletion power. (b) A series of STED-DLW nanowires with 45 nm width. W: scanning direction, E: laser polarization direction, Scale bar 200 nm.
Secondly, in abstract and conclusion, the authors mention that “We expect that our approach, in particular, the fabrication of nanowires below 50 nm could be a prospective fabrication method for nanostructured components and devices.” or “Our approach, capable of writing stable and continuous nanowires below 50 nm, could offer a promising high-resolution manufacturing method for nanoscale devices and components.”
Which nanoscale devices and components ?
Because, it is clear that this twin-oval focusing spot can only reduce the focusing spot side in one direction. By scanning in one direction (a straight line), we can obtain a narrow straight line. This is not possible to obtain a curved line or a 3D structure with same small size ! So, which nanoscale devices and components ? Only a series of straight nanowires can be realized by this technique.
The authors should remove these sentences.
Response: We thank the reviewer for pointing this out, while we have different opinion. Our method has limitation of size reduction in one direction, it still capable of fabrication some 2D and 3D nano-structures that have varies of application, such as Nano-brush for Disc storage devices (Ref. 3), nano-grating for cell filtration (Ref. 4) and thin-wall for micro-fluidic devices (Ref. 5). We have added line to indicate the application of our method in Introduction section in p. 2 as: “ Our approach, capable of writing stable nanowires and structures that sub-diffraction limit, could be a convenient nanofabrication method for nano-grating for cell filtration [15], thin-wall for micro-fluidic devices [18]and Nano-brush for Disc storage devices [33], etc. ”
Similarly, in the section 3, the authors write: “Here, we simply write straight nanowires instead of 3D nanostructures on the cover slice substrate, since the widths from single nanowires have a higher reliability than those feature size from 3D nanostructures.”
That sentence is not correct, because no 3D structure can be fabricated by this method! as explained above.
Response: We thank the reviewer for pointing this out, the describing on 3D nanostructures has been deleted, and the line is modified to: “Straight nanowires were written to explore the effect of photoinitiators.”
The result shown in Figure 6 is not convinced. In figure 6a, I don’t see any reduction of the wire size when the STED beam in on (it is very clear in Figure 5). Simply a reduction of the height of the wire, as shown in Figure 6c. So the width is not reduced? If the STED works for 30 mW of power (continue from the result of Figure 5h), we should first see a reduction of the wire width, even this “small wire” will be collapsed and washed out during the development process. Here, clearly see in Figure 6a, that the width is the same. The authors should check for the result at higher STED laser power and correct this result.
Response: We thank the reviewer for pointing this out. There do have width reduction with 30 mW depletion power. Some photoresists remain on the surface of the substrate when their top part are washed away, which’s height is pretty low and not easily divided with the residual nanowire. The width reduction is quite obvious in the AFM image. Since the previous image in not very clear, a new SEM image of 30 mW depletion power has been uploaded to the revised manuscript in Figure 7a.
Figure 7. Scanning electron micrograph of STED direct laser writing nanowires. (a) 11mW two-photon excitation power, 30 mW depletion power.
Finally, the authors should correct the term “one-photon” and “two-photon” because there are many times in the text, it is written as “two photon”.
Response: We thank the reviewer for pointing this out, and we have corrected this mistake in the revised manuscript.
The manuscript requires thus a major revision
Response: A major revision has been made to the manuscript according to the comments of the reviewer. Please check the Reponses and the revision, thanks again for your brilliant suggestion and we look forward to hear from you.
Stöhr, R.J.; Kolesov, R.; Xia, K.; Wrachtrup, J. All-Optical High-Resolution Nanopatterning and 3D Suspending of Graphene. ACS Nano 2011, 5, 5141-5150. Harke, B.; Bianchini, P.; Brandi, F.; Diaspro, A. Photopolymerization Inhibition Dynamics for Sub-Diffraction Direct Laser Writing Lithography. ChemPhysChem 2012, 13, 1429 – 1434 Mueller, P.; Zieger, M. M.; Richter, B.; Quick, A. S.; Fischer, J.; Mueller, J. B.; Zhou, L.; Nienhaus, G. U.; Bastmeyer, M.; Barner-Kowollik, C.; Wegener, M. Molecular Switch for Sub-Diffraction Laser Lithography by Photoenol Intermediate-State Cis-Trans Isomerization. ACS Nano 2017, 11, 6396– 6403. Au, T.H.; Trinh, D.T.; Tong, Q.C.; Do, D.B.; Nguyen,D.P.; Phan, M.H.; Lai, N.D. Direct Laser Writing of Magneto-Photonic Sub-Microstructures for Prospective Applications in Biomedical Engineering. Nanomaterials 2017, 7(5), 105. Sugioka, K.; Xu, J.; Wu, D.; Hanada, Y.; Wang, Z. K.; Cheng, Y.; Midorikawa, K. Femtosecond laser 3D micromachining: a powerful tool for the fabrication of microfluidic, optofluidic, and electrofluidic devices based on glass. Lab Chip 2014, 14, 3447-3458.

Reviewer 2 Report
Comments:
Feature size is a significant point that decides the capability of laser nano-printing and STED technology is a promising method that has been employed for the delicate fabrication of nanostructures below the light diffraction limit. This manuscript presents experimental investigations on fabrication of nanostructures with feature size of tens of nanometers by STED technology. The effects of photoinitiators and laser powers on linewidth are explored and feature size down to 45 nm is claimed with twin-oval depletion focus. However, there is rarely new information in this manuscript including the methods, materials, and even the achieved line width. Following are the main concerns of the contents:
As the authors mentioned in the introduction, feature size of 55 nm (Opt. Exp. 2013, 21, 10831-10840) and 53 nm (Opt. Exp. 2013, 7, 2538-2559) have already been achieved in previous reports. Even single lines with 9 nm feature size has been realized. So, 45 nm linewidth is not a very impressive or competitive results. One thing the manuscript pointed is the twin-oval depletion focus instead of vortex depletion focus, and the rod shaped effective focus which is plate shaped focus in fact. However, the twin-oval depletion focus has also been reported for several times in STED lithography (ACS Nano 2017, 11, 6396−6403; Chem. Mater. 2019, 31, 1966−1972), and even line width of 31.2 nm has been achieved. The minimun linewidth realized is 45 nm. However, there is no information of how the linewidth is measured in detail and what is the standard error of the measurement. In my point of view, considering the error of measurement by SEM images, the feature size of 45 nm, 53 nm and 55 nm are in the same range, which is the typical size achieved with DETC plus PETA formula and STED lithography. One thing the authors should pay attention to is that the lines should attached to the substrate to eliminate the effect of shrinkage after development. And only in this way, the real linewidth can be extracted. This means the focus should start from the inside of the substrate. Thus, how much of the effective focus is above the substrate may have a dramatic effect on the final size of fabricated lines. So, the 45 nm feature size may be attributed to not only the depletion from the depletion beam, but also from the incomplete focus above the substrate. One clue is that the line height from excitation laser only is measure to be 60 nm in Fig. 6(c), which should be larger than the line width (120 nm) since the focus has a elliptical geometry with 2-3 times size in the z direction.Author Response
Responses to Reviewer’ Comments of micromachines-610158
The authors would like to thank the reviewer for the insightful comments. The following are detailed responses to the reviewer’s comments.
Feature size is a significant point that decides the capability of laser nano-printing and STED technology is a promising method that has been employed for the delicate fabrication of nanostructures below the light diffraction limit. This manuscript presents experimental investigations on fabrication of nanostructures with feature size of tens of nanometers by STED technology. The effects of photoinitiators and laser powers on linewidth are explored and feature size down to 45 nm is claimed with twin-oval depletion focus. However, there is rarely new information in this manuscript including the methods, materials, and even the achieved line width. Following are the main concerns of the contents:
As the authors mentioned in the introduction, feature size of 55 nm (Opt. Exp. 2013, 21, 10831-10840) and 53 nm (Opt. Exp. 2013, 7, 2538-2559) have already been achieved in previous reports. Even single lines with 9 nm feature size has been realized. So, 45 nm linewidth is not a very impressive or competitive results. One thing the manuscript pointed is the twin-oval depletion focus instead of vortex depletion focus, and the rod shaped effective focus which is plate shaped focus in fact.
However, the twin-oval depletion focus has also been reported for several times in STED lithography (ACS Nano 2017, 11, 6396−6403; Chem. Mater. 2019, 31, 1966−1972), and even line width of 31.2 nm has been achieved. The minimun linewidth realized is 45 nm. However, there is no information of how the linewidth is measured in detail and what is the standard error of the measurement.
In my point of view, considering the error of measurement by SEM images, the feature size of 45 nm, 53 nm and 55 nm are in the same range, which is the typical size achieved with DETC plus PETA formula and STED lithography.
One thing the authors should pay attention to is that the lines should attached to the substrate to eliminate the effect of shrinkage after development. And only in this way, the real linewidth can be extracted. This means the focus should start from the inside of the substrate. Thus, how much of the effective focus is above the substrate may have a dramatic effect on the final size of fabricated lines. So, the 45 nm feature size may be attributed to not only the depletion from the depletion beam, but also from the incomplete focus above the substrate. One clue is that the line height from excitation laser only is measure to be 60 nm in Fig. 6(c), which should be larger than the line width (120 nm) since the focus has a elliptical geometry with 2-3 times size in the z direction.
Response: We thank the reviewer for bringing these papers to our attention. We aim at fabricating stable nanowires below 50 nm with STED lithography. For STED with a donut-shaped depletion focus, the minimum linewidth or feature size is 55 nm (Ref. 1) and 53 nm (Ref. 2). The 9 nm feature size nanowire (Ref. 3) is generated by the gap between excitation and depletion focus, while it is quite short and not very stable.
For STED with twin-oval depletion focus, since the excitation focus is not totally encircle by the depletion, theoretically, the width of STED-DLW nanowires can reach 10 nm or even several nanometers. The minimum width of nanowire obtained by Mueller et al. (Ref. 4) is 60 nm. Then, they pushed this number to 31.2 nm with a new photoresist (Ref. 5). While, these lines are discontinuous and full of burrs, and even nanowires with 50 nm width has the same problem. And our method capable of fabricating stable nanowires bellowing 50 nm. An image of multiply nanowires with 45 nm width has been added to the revised manuscript in Figure 7b, which means that 45 nm width is stable and repeatable. A line to describe the details of this figure has been added in p. 6, as: “Figure 7b is a series of STED-DLW nanowires with 45 nm width, and the length of these nanowires is about 1 μm.”
Figure 7. Scanning electron micrograph of STED direct laser writing nanowires. (b) A series of STED-DLW nanowires with 45 nm width. W: scanning direction, E: laser polarization direction, Scale bar 200 nm.
Regarding the comments on the measurement of linewidth, we use an imaging software combining with the SEM images. A series of these lines (at least 5), the average of them was post on the SEM image. We have added description on width measurement in Characterization section p.2 as: “The width of nanowires were measured with the combination of an imaging software and SEM images.” And p.5 as: “These linewidth is averaged by a series of (at least 5) nanowires with the same parameters to reduce error.”
For the result of 45 nm, no significant difference with the previous work of 55 nm, which is confined by the global connectivity between the polymer chains. A new monomers with higher crosslinking density and mechanical strength can greatly broaden the processing of linewidth and resolution. We will search for new monomer, reduce the limit of linewidth and conduct sub-diffraction limit 3D structures in the future.
Regarding the comments focus position at z direction in STED-DLW process, we have performed AFM scanning of these nanowires. The result has been shown below (Figure 6 in the revised manuscript). The one-beam DLW nanowire has a height of 72 nm. The height of STED-DLW nanowires is 55 nm, 47 nm, 39 nm and 32 nm, respectively. With the increasing of depletion power, the height and width of STED-DLW nanowires decrease simultaneously. The detailed description can be found in the revised manuscript. The height of these nanowires is smaller than their width, but much bigger than half of it. There is still a part of nanowires under the substrate. While, if we move the focus up (z=+0.1 μm or 0.2 μm), the nanowire will be washed during the developing process. So, our method can find application to fabricate semi-circular nanowires.
Figure 6. AFM images and height profile of the four nanowires, with 11mW two-photon excitation power and depletion power from left to right 10 mW, 15mW, 20mW, and 25mW respectively.
Wollhofen, R.; Katzmann, J.; Hrelescu, C.; Jacak, J.; Klar, T. A. 120 nm resolution and 55 nm structure size in STED-lithography. Opt. Exp. 2013, 21, 10831-10840. Wollhofen, R.; Buchegger, B.; Eder, C.; Jacak, J.; Kreutzer, J.; Klar, T. A. Functional photoresists for sub-diffraction stimulated emission depletion lithography. Opt. Mater. Express 2017, 7, 2538-2559. Gan Z. S., Cao Y. Y., Evans R. A.; Gu M. Three-dimensional deep sub-diffraction optical beam lithography with 9 nm feature size. Nat. Commun. 2013, 4, 2061. Mueller, P.; Zieger, M. M.; Richter, B.; Quick, A. S.; Fischer, J.; Mueller, J. B.; Zhou, L.; Nienhaus, G. U.; Bastmeyer, M.; Barner-Kowollik, C. Molecular Switch for Sub-Diffraction Laser Lithography by Photoenol Intermediate-State Cis-Trans Isomerization. ACS Nano 2017, 11, 6396– 6403. Müller, P.; Müller, R.; Hammer, L.; Barner-Kowollik, C.; Wegener, M.; Blasco, E. STED-inspired Laser Lithography Based on Photoswitchable Spirothiopyran Moieties. Chem. Mater. 2019, 31, 1966–1972.

Reviewer 3 Report
The manuscript by He et al. describes STED DLW photolithography using ITX and DETC photoinitators and an atypical phase mask. They demonstrate a reduction of feature sizes using the STED principle with both photoinitiators but since the DETC showed a larger decrease they used that system to show the creation of 45 nm linewidth features. The work is good and I think will be of interest to the readers of micromachines but it only modestly advances the field of STED-DLW. These initiators have been used before by leaders in the field and so mainly what is new is the phase mask that was used. Importantly, the language is a bit unclear about the orientation of the phase mask. It would seem based on the figure 1c that the mask results in “twin oval” mode beam pattern in the x-y plane (as opposed to the x-z plane), but the axes are not labeled in the figure. If it is in the x-y plane then the depletion beam’s affect would be different if the line were scanned along the x-axis or y-axis, that is if the path were parallel or perpendicular to the “rod shape” effective focus. I suspect this is the case because in the description of the results they say the path is bottom-up (although I think it would be clearly to say parallel to the y-axis, because up and down have no meaning). If this is correct it severely limits the utility of this phase mask since the reduction in feature sizes only occurs in one direction. Despite this, I think the manuscript is interesting and will be of interest to the readers but I suggest publishing it with some major revisions to address the following concerns:
1) Show what happens when the lines are written parallel and perpendicular to the phase mask twin ovals. I suspect the linewidth will only get smaller for one of these directions, which is ok, but the authors should not ignore this fact.
2) The numerical results have no statistical analysis. The authors need to cite how many lines were measured (n), and what was the average and standard deviation of these measurements. I would recommend that n >10, preferable for different samples and different experimenters, for any number cited to lend some credibility to their claims.
3) I am convinced based on the data that the STED beam reduces the linewidths but I am not convinced that the lines are drawn with the center of the voxel at the surface (z=0). To prove this a z-scan through the surface needs to be performed which will give the height of the lines and not just their width. My real concern is that the lines shown in the paper are just the tip of a wider line that is partially submerged in the substrate and if the user missed the surface by a fraction of micron (e.g. z=+0.2 um instead of z = 0 um) the line width could grow significantly. Of course part of this concern will be answered by #2 with repeated trials. Also, the authors should comment on 3-D patterning with this system. Can you write with these high resolution lines to build 3-D objects?
4) There are several general grammar corrections and a lack of attention to detail. There are several minor English grammatical mistakes and I would suggest the manuscript be edited by a professional copy editor. There are several instances of small mistakes, like where a number and its unit are not separated (e.g. 532nm should be 532 nm) or the acronym for the phase plate is EPM and EFM, but it should be one or the other.
5) The comment at the end of the introduction about the work by Gan et al. only being able to generate periodic adjacent lines is incorrect. In that work they demonstrated 9 nm linewidth and also went on to characterize the resolution, or space between lines, which I would recommend these authors do as well for their work. Please correct this.
6) In the line that says the power was measured “in front of the microscope lens” the language is ambiguous; this could mean before or after the objective, please specify.
Author Response
Responses to Reviewer’ Comments of micromachines-610158
The authors would like to thank the reviewer for the insightful comments. The following are detailed responses to the reviewer’s comments.
The manuscript by He et al. describes STED DLW photolithography using ITX and DETC photoinitators and an atypical phase mask. They demonstrate a reduction of feature sizes using the STED principle with both photoinitiators but since the DETC showed a larger decrease they used that system to show the creation of 45 nm linewidth features. The work is good and I think will be of interest to the readers of micromachines but it only modestly advances the field of STED-DLW. These initiators have been used before by leaders in the field and so mainly what is new is the phase mask that was used.
Importantly, the language is a bit unclear about the orientation of the phase mask. It would seem based on the figure 1c that the mask results in “twin oval” mode beam pattern in the x-y plane (as opposed to the x-z plane), but the axes are not labeled in the figure. If it is in the x-y plane then the depletion beam’s affect would be different if the line were scanned along the x-axis or y-axis, that is if the path were parallel or perpendicular to the “rod shape” effective focus. I suspect this is the case because in the description of the results they say the path is bottom-up (although I think it would be clearly to say parallel to the y-axis, because up and down have no meaning). If this is correct it severely limits the utility of this phase mask since the reduction in feature sizes only occurs in one direction. Despite this, I think the manuscript is interesting and will be of interest to the readers but I suggest publishing it with some major revisions to address the following concerns:
1) Show what happens when the lines are written parallel and perpendicular to the phase mask twin ovals. I suspect the linewidth will only get smaller for one of these directions, which is ok, but the authors should not ignore this fact.
Response: We thank the reviewer for point this out. The polarization of two laser beams do have effects to our experiment. Nanowires have width reduction only in one direction, which’s writing direction parallel to the polarization direction. For the writing direction perpendicular to the laser polarization, these is no size reduction, as shown in Figure A. We test the effect at the beginning of this research. The power we using is pretty big at that time, 25 mW excitation power and 60 mW depletion power. The one-beam DLW nanowires has a width of 360 nm, same with the value obtained from two-beam STED-DLW nanowire that writing direction perpendicular to the laser polarization direction. Since we aim at getting the thinnest nanowire, nanowires without size reduction do not show up in this paper. Accordingly, the following line is added to Result and Discussion section in p. 4 as “For the two-beam STED-direction nanowire, There is no size reduction when the writing direction perpendicular to laser polarization direction.”
The label of x-y plane has been added to Figure 1b and 1c, and the drawing vision of excitation and depletion laser focus is also replaced by their intensity distribution. The writing direction of nanowires in Figure 3 to Figure 6 has been modified as “along y direction, parallel to the polarization direction.” The following line is added to Method section in p. 2 as “A beam splitter was inserted into both the two laser beams before the objective lens to make them linearly polarized, and the 100x oil immersion objective lens (Apo TIRE, Nikon, Japan) has a NA of 1.49.”
Figure A. STED-DLW nanowire with 25 mW excitation power and 60 mW depletion power. (a) Writing direction parallel to the laser polarization direction, (b) Writing direction perpendicular to the laser polarization direction, W: scanning direction, E: laser polarization direction.
Figure 1. (a) Schematic diagram of STED direct laser writing. (b) Creation of the twin-oval depletion focus by inserting a half-space edge phase mask into the depletion focus. (c) Formation of rod-shaped effective focus spot, which is a combination of Gaussian excitation focus coincides with twin-oval depletion focus, Scale bar 200 nm.
2) The numerical results have no statistical analysis. The authors need to cite how many lines were measured (n), and what was the average and standard deviation of these measurements. I would recommend that n >10, preferable for different samples and different experimenters, for any number cited to lend some credibility to their claims.
Response: We thank the reviewer for pointing this out. We use an imaging software combining with the SEM image to measure the width of nanowires. We measured a series of these lines, the average of them was post on the SEM image. We have added description on width measurement in Characterization section p.2 as: “The width of nanowires were measured with the combination of an imaging software and SEM images.” And p.5 as: “These linewidth is averaged by a series of (at least 5) nanowires with the same parameters to reduce error.”
3) I am convinced based on the data that the STED beam reduces the linewidths but I am not convinced that the lines are drawn with the center of the voxel at the surface (z=0). To prove this a z-scan through the surface needs to be performed which will give the height of the lines and not just their width. My real concern is that the lines shown in the paper are just the tip of a wider line that is partially submerged in the substrate and if the user missed the surface by a fraction of micron (e.g. z=+0.2 um instead of z = 0 um) the line width could grow significantly. Of course part of this concern will be answered by #2 with repeated trials. Also, the authors should comment on 3-D patterning with this system. Can you write with these high resolution lines to build 3-D objects?
Response: As the suggestion of the reviewer, a z-scan of nanowires in Figure 5 has been performed to explore the focus position in z direction. The results have been shown as AFM image and height profile below (Figure 6 in the revised manuscript). The DLW nanowire has a height of 72 nm. The STED-DLW nanowire has a height of 55 nm, 47 nm, 39 nm and 32 nm, respectively. With the increasing of depletion power, the height and width of STED-DLW nanowires decrease simultaneously. The detailed description of AFM image can be found in the revised manuscript. The height of these nanowires is much bigger than half of the width. Which means there is still a part of nanowires under the substrate. But, obviously, these nanowires are not just the tip of a wider line. While, if we move the focus up (z=+0.1 μm or 0.2 μm), the nanowire will be washed during the developing process. So our method can find application to fabricate semi-circular nanowires.
We could make a 3D nanostructures, while the limitation of size reduction only in one direction restrict its applications in some kind. We concentrate on getting the thinner nanowire in this work, and will conduct 3D nanostructures with sub-diffraction limit in the near future.
Figure 6. AFM images and height profile of the four nanowires, with 11mW two-photon excitation power and depletion power from left to right 10 mW, 15mW, 20mW, and 25mW respectively.
4) There are several general grammar corrections and a lack of attention to detail. There are several minor English grammatical mistakes and I would suggest the manuscript be edited by a professional copy editor. There are several instances of small mistakes, like where a number and its unit are not separated (e.g. 532nm should be 532 nm) or the acronym for the phase plate is EPM and EFM, but it should be one or the other.
Response: We thank the reviewer for careful reading and pointing out these English grammatical mistakes, we have corrected these errors as: “532nm” ==> “532 nm”, and abbreviated Edge Phase Mask as EPM. We also checked the whole document to correct other errors and typos we can find.
5) The comment at the end of the introduction about the work by Gan et al. only being able to generate periodic adjacent lines is incorrect. In that work they demonstrated 9 nm linewidth and also went on to characterize the resolution, or space between lines, which I would recommend these authors do as well for their work. Please correct this.
Response: We thank the reviewer for point this out. In the revised manuscript, we have modified the portion describing Gan’s work in the introduction section in p. 2 as “The 9 nm feature size nanowire is generated by the gap between excitation and depletion focus, while it is quite short and not very stable.”
6) In the line that says the power was measured “in front of the microscope lens” the language is ambiguous; this could mean before or after the objective, please specify.
Response: We thank the reviewer for point out this mistake. In the revised manuscript p4, line 129, we have replaced “in front of the microscope lens” with “before the objective lens”.

Round 2
Reviewer 1 Report
I receive all three responses to the three referees, and I found that the authors made a lot of improvement of the manuscript.
However, there is a big problem in the response related to 3D fabrication. The authors agree with me that this technique is only applied for fabrication of 1D straight lines structures, not possible for 2D and not for 3D ! One of the referees also has the same comment like me. But in the answer to another referee, the authors confirm that the technique can be applied for 3D structure, and they are doing such demonstration !
In that case, I propose that the authors show this 3D structures, with very good resolution of 45 nm !
Also, the authors made a series of lines as I recommended. But these lines are not "stables" and not "long enough" to be confirmed as "very stable" as in the manuscript. As commented by another referee, these lines have different sizes (and even not straight) with a variation of at least 10 nm ! The size of 45 nm is not a relible value to say that is better than 50 nm (of beautiful structure) in other papers. I don't agree with the authors to continue to mention that the structures fabricated by other groups are not stable, and the structures shown in manuscript are stable !
Even the authors made a lot of efforts to show that this work is a "new" and "useful", I think that the results do not support that claim.
Author Response
Responses to Reviewer’ Comments of micromachines-610158
The authors would like to thank the reviewer for the insightful comments. The following are detailed responses to the reviewer’s comments.
Reviewer #1
I receive all three responses to the three referees, and I found that the authors made a lot of improvement of the manuscript.
However, there is a big problem in the response related to 3D fabrication. The authors agree with me that this technique is only applied for fabrication of 1D straight lines structures, not possible for 2D and not for 3D! One of the referees also has the same comment like me. But in the answer to another referee, the authors confirm that the technique can be applied for 3D structure, and they are doing such demonstration!
In that case, I propose that the authors show this 3D structures, with very good resolution of 45 nm!
Response: Our method do have limitation of size reduction only in one direction, while we can shift this reducing direction by rotating laser polarization direction and edge phase mask. In this way, sub-diffraction limit 3D nanostructures like grating, waveguide and lattice can be fabricated. Of course, this function needs a lot of complex beam alignments and real 3D nanostructures with curves cannot be achieved. It is our fault to misunderstand the reviewer’s comment, and we thank the reviewer for pointing out our limitation. The describing sentences on 3D nanostructures in Abstract, Introduction and Conclusion section has been deleted, and the following changes are made:
“We expect that our approach, in particular, the fabrication of nanowires below 50 nm could be a prospective fabrication method for nanostructured components and devices.” in Abstract section p.1 is changed to: “Our result accelerates the progress of achievable linewidth reduction in STED direct laser writing.”
“A smaller line width can be achieved by applying higher depletion power. Our approach, capable of writing stable nanowires and structures that sub-diffraction limit, could be a convenient nanofabrication method for nano-grating for cell filtration [15], thin-wall for micro-fluidic devices [18] and Nano-brush for Disc storage devices [33], etc.” in Introduction section p.2 has been changed to “A series of nanowires with 45 nm width is fabricated to verify the stability of our method, and a higher depletion power is applied to explore the smaller nanowire width.”
“A smaller line width can be achieved by applying higher depletion power, while the nanowires may not stable or continuous. Our approach, capable of writing stable and continuous nanowires below 50 nm, could offer a promising high-resolution manufacturing method for nanoscale devices and components.” in Conclusion section p.7 has been changed to “A series of nanowires with 45 nm width were fabricated to verify the stability of our method, and a thinner nanowire can be achieved with a higher depletion power and monomer of higher crosslinking density. Our method, capable of writing stable and continuous nanowires below 50 nm, has accelerated the progress of precision enhancement in optical nanofabrication.”
Also, the authors made a series of lines as I recommended. But these lines are not "stables" and not "long enough" to be confirmed as "very stable" as in the manuscript. As commented by another referee, these lines have different sizes (and even not straight) with a variation of at least 10 nm! The size of 45 nm is not a reliable value to say that is better than 50 nm (of beautiful structure) in other papers. I don't agree with the authors to continue to mention that the structures fabricated by other groups are not stable, and the structures shown in manuscript are stable!
Even the authors made a lot of efforts to show that this work is a "new" and "useful", I think that the results do not support that claim.
Response: We thank the reviewer for pointing out this confusion. The emphasis of “stable” is for nanowires below 50 nm. Klar et al. use STED-DLW with a donut-shaped depletion focus to fabricate 2D and 3D sub-diffraction nanostructures (Ref. 1-3). These nanowires and lines in 3D structures are stable and continuous. While, the minimum linewidth or feature size in their work is 55 nm (Ref. 1) and 53 nm (Ref. 2-3). Gan et al. use the gap between excitation and donut depletion focus in STED lithography, and achieve single and two adjacent lines with 9 nm feature size (Ref. 4). While, these line structures below 50 nm are quite short and not very stable. We aim at writing sub-50 nm nanowires with stable and continuous structure, and the thinnest nanowires we obtained has a width of 45 nm. In the revised manuscript, we modified this confusion in Introduction section in p. 1, as: “In conventional STED-DLW, normally a donut-shaped depletion focus was used to fabricate stable nanowires and 3D sub-diffraction nanostructures. Since the excitation focus is totally encircled by the depletion focus and excessive depletion power can also excite the photoresist molecules, the minimum width or feature size in lateral direction is restricted to 55 nm [30] and 53 nm [28, 31]. Gan et al. use the gap between excitation and donut depletion focus in STED lithography, and achieve single and two adjacent lines with 9 nm feature size [35]. While, these line structures below 50 nm are quite short and not very stable. ” For the comments of cannot be confirmed as “very stable”, we have removed this description on our nanowires in p. 6.
Regarding the comments of “these lines have different sizes (and even not straight)”, this is due to the uneven substrate surface and developing process. Just as the reviewers point out, our STED-DLW method is very sensitive to the focus position in z direction. A shift of z=-0.05 μm will lead to the disappearance of STED nanowires, and a shift of z=+0.05 μm or less will result with the washed away or fell down of nanowires. This explanation is further demonstrated by the fact that the top half of the fourth STED nanowire is much thicker than the bottom half of the fourth STED nanowire in Figure 7b (as shown below). The height of nanowires should be larger than the line width since the focus has an elliptical geometry with 2-3 times size in the z direction. Excluding the height of fourth STED nanowires, the width of STED nanowire may be deviated, but the error should be less than 5nm. And, 45 nm is the average width of these STED nanowires. On the other hand, we will search for a smoother substrate and level it before lithography experiment in our future work.
Figure 7. Scanning electron micrograph of STED direct laser writing nanowires. (a) 11mW two-photon excitation power, 30 mW depletion power. (b) A series of STED-DLW nanowires with 45 nm width. W: scanning direction, E: laser polarization direction, Scale bar 200 nm.
Thanks so much for your comments on our manuscript. We look forward to hearing from you regarding our submission. We would be glad to respond to any further questions and comments that you may have.
Wollhofen, R.; Katzmann, J.; Hrelescu, C.; Jacak, J.; Klar, T. A. 120 nm resolution and 55 nm structure size in STED-lithography. Opt. Exp. 2013, 21, 10831-10840. Klar, T. A.; Wollhofen R.; Jacak. J. Sub-Abbe resolution: from STED microscopy to STED lithography. Phys. Scr. 2014, T162, 014049. Wollhofen, R.; Buchegger, B.; Eder, C.; Jacak, J.; Kreutzer, J.; Klar, T. A. Functional photoresists for sub-diffraction stimulated emission depletion lithography. Opt. Mater. Express 2017, 7, 2538-2559. Gan Z. S., Cao Y. Y., Evans R. A.; Gu M. Three-dimensional deep sub-diffraction optical beam lithography with 9 nm feature size. Nat. Commun. 2013, 4, 2061.

Reviewer 2 Report
The revised manuscript satisfies standard of publication in Micromachines although some points in the original comments were not completely fixed. I think it would be of general interest to the readers.
Author Response
Responses to Reviewer’ Comments of micromachines-610158
The revised manuscript satisfies standard of publication in Micromachines although some points in the original comments were not completely fixed. I think it would be of general interest to the readers.
Response: Thanks so much for the recommendation of this manuscript. We also appreciate the reviewer for the valuable suggestions and questions. Some suggestions or comments may not be fully fixed due to lack of time or different opinion. We will continue to work on this manuscript, and make it a better article.

Reviewer 3 Report
The authors have addressed my concerns. I recommend the manuscript be published.
Author Response
Responses to Reviewer’ Comments of micromachines-610158
The authors have addressed my concerns. I recommend the manuscript be published.
Response: Thanks the reviewer for recommending this manuscript. We also appreciate very much for the insightful advices that reviewer has given, which have great benefits for the improvement of this paper and our future research.

Round 3
Reviewer 1 Report
Thank you for your effort in response, even there is still a lot of work to do.